# An HBase-Based Optimization Model for Distributed Medical Data Storage and Retrieval

Chengzhang Zhu [1,2,3,4], Zixi Liu [1,3,4], Beiji Zou [1,3,4], Yalong Xiao [1,2,3,4,*], Meng Zeng [1,3,4], Han Wang [1,3,4] and Ziang Fan [2,3]

1 Department of Computer Science, Central South University, Changsha 410083, China
2 Department of Literature and Journalism, Central South University, Changsha 410083, China
3 Mobile Medical Ministry of Education-China Mobile Joint Laboratory, Changsha 410083, China
4 Machine Vision and Smart Medical Engineering Technology Center, Changsha 410083, China
* Correspondence: ylxiao@csu.edu.cn

**Abstract:** In medical services, the amount of data generated by medical devices is increasing explosively, and access to medical data is also put forward with higher requirements. Although HBase-based medical data storage solutions exist, they cannot meet the needs of fast locating and diversified access to medical data. In order to improve the retrieval speed, the recognition model S-TCR and the dynamic management algorithm SL-TCR, based on the behavior characteristics of access, were proposed to identify the frequently accessed hot data and dynamically manage the data storage medium as to maximize the system access performance. In order to improve the search performance of keys, an optimized secondary index strategy was proposed to reduce I/O overhead and optimize the search performance of non-primary key indexes. Comparative experiments were conducted on real medical data sets. The experimental results show that the optimized retrieval model can meet the needs of hot data access and diversified medical data retrieval.

**Keywords:** big medical data; HBase; retrieval optimization; hot data; secondary index

## 1. Introduction

With the rapid development of medical information technology, medical treatment and medical research are stepping into the era of big data [1]. According to the National Hospital Information Construction Standard and Specification (Trial) issued by the National Health Commission in 2018, the data storage part of the infrastructure construction of the information platform should realize the unified storage, processing, and management of the platform data [2]. Massive medical big data contains a great value and can provide data support for remote consultation, medical consultation, medication recommendation, daily health care, and other services. Therefore, it is of great significance to construct unified storage and retrieval of medical data. However, big medical data are characterized by their large scale, diverse structure, fast growth, and multiple modes, which brings great challenges to unified storage, retrieval, and management. The traditional relational storage system can no longer guarantee low-cost, large-capacity storage and fast retrieval of massive medical data [3–5]. Due to the development of emerging distributed systems, HBase [6–8], a distributed column database, has become a mainstream medical data storage model that meets the goal of low-cost and high-capacity storage of massive medical data. Considering the specificity of the medical service industry, fast response and diversified retrieval have become the necessary design objectives for the medical data storage model because of the large amount of data required to support clinical decision-making and other tasks [9,10]. Hence, how to meet the need for the fast and diverse retrieval for HBase-based medical data storage models is an urgent problem.

In order to achieve the fast and diverse retrieval of medical data, this paper is optimized from the following two aspects. First, aiming at meeting the requirements of the

rapid retrieval of data, data that are frequently accessed are called hot data, while data that are occasionally accessed or not accessed are called cold data [11–14]. Considering that modern computers use a hybrid storage architecture, the closer the storage media is to the CPU, the faster the access speed, the smaller the capacity, and the higher the cost, which is used to balance the cost and performance of storage media. We designed a data dynamic management model to realize hot data identification and storage media management. By using this model, the hot data were stored on the high-speed device (i.e., the Hot Area) while the cold data were stored on the low-speed device (i.e., the Cold Area), thus meeting the performance requirements of the frequent interactions of hot medical data. Second, a secondary index was constructed to support the retrieval of non-primary keys and to meet the needs of medical data diversity queries. Inverted indexing was used to build a secondary index with the core idea of storing a map from the keys to the corresponding primary key [15–18]. This solution was simple and easy to implement, but multiple I/O operations may lead to high time overhead and even performance bottlenecks. Therefore, in the current study, Bloom Filter (BF) and index position optimization methods were adopted to reduce the overhead of I/O and optimize the secondary index.

In conclusion, in order to optimize the storage system's performance, a model for dynamically identifying and managing hot and cold data and an optimized secondary index optimization strategy were proposed in this study. The main contributions of this paper are as follows:

(1) A data temperature recognition method S-TCR and a data management algorithm SL-TCR were proposed to manage medical data dynamically;
(2) An optimized secondary indexing strategy was proposed to improve the speed of medical data diversity queries;
(3) The feasibility and efficiency of the proposed model were verified by experiments on real medical data sets.

This paper is organized as follows: (1) the background knowledge was summarized in Section 2; (2) the dynamic data management model and the optimization strategy of secondary index retrieval were introduced in Section 3; (3) the experimental setup and results were given in Section 4; and (4) this paper was summarized in Section 5.

## 2. Background Knowledge

In this section, the HBase database was first briefly introduced. Then, the hot and cold data management algorithms were studied. Finally, the idea of a secondary index was introduced.

### 2.1. HBase Database

HBase is a column-oriented database running on a Hadoop cluster. Hadoop is a distributed cluster deployed on multiple machines. A Hadoop cluster connects multiple servers through a network to provide external storage services as a whole [19,20]. Hadoop Distributed File System (HDFS) can store and read massive amounts of data in a distributed manner and provide high-throughput data access. Therefore, Hadoop is well suited for building a mass data storage platform [21,22].

For application requests that require random data to be read, data can be chosen to be stored in HBase. HBase stores underlying data in the HDFS to ensure data reliability. As shown in Figure 1, the HBase cluster consists of Master, RegionServer, Region, and Zookeeper components [23]. The Master is the primary server of the HBase cluster and allocates RegionServers to regions. The RegionServer component is responsible for providing write, delete, and search services to clients [24]. The Region component is a sub-table divided by RowKey. It is the smallest storage and processing unit in HBase. The RowKey is the unique identifier for the HBase record [25]. The ZooKeeper provides application coordination services for the HBase cluster, detects and clears failed Masters, and elects a new active Master.

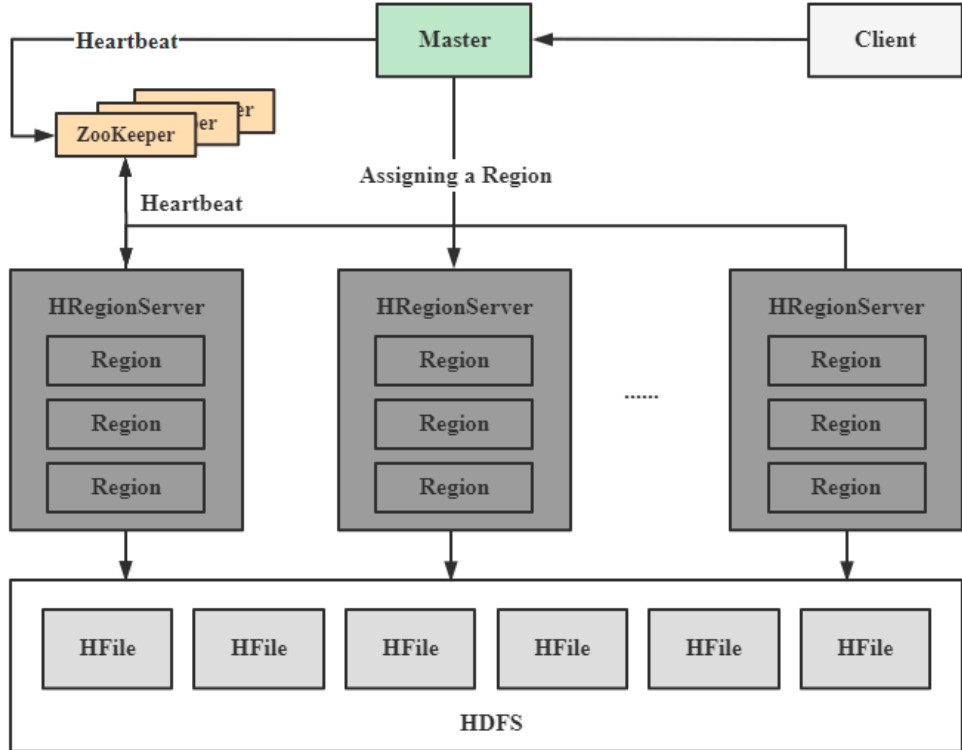

**Figure 1.** The structure of HBase.

### 2.2. Hot and Cold Data Management Algorithms

In order to utilize the Hot Area more effectively and ensure its hit rate, the Hot Area and Cold Area should have efficient management algorithms, which are commonly used as follows:

#### 2.2.1. LRU

The LRU (Least Recently Used) [26] algorithm manages data in the Hot Area according to the access time of historical records. The algorithm is shown in Figure 2, where the recently accessed data are more likely to be accessed in the future. It is simple and easy to implement, but the hit rate is low for random and periodic accesses, i.e., the Hot Area is heavily contaminated.

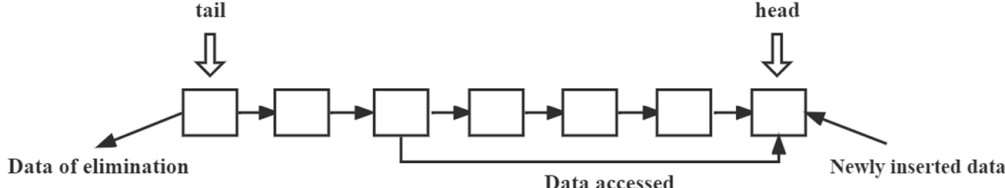

**Figure 2.** The idea of the LRU.

#### 2.2.2. LFU

The LFU (Least Frequently Used) algorithm [27] manages the data in the Hot Area according to the access frequency of historical records. The idea of the algorithm is shown in Figure 3. It uses a counter to count the number of accesses to each object, and when a replacement occurs only the least accessed data needs to be moved out of the Hot Area. However, it does not consider the object access interval and object size, resulting in dead hot data wrote to storage space and the pollution of the Hot Area.

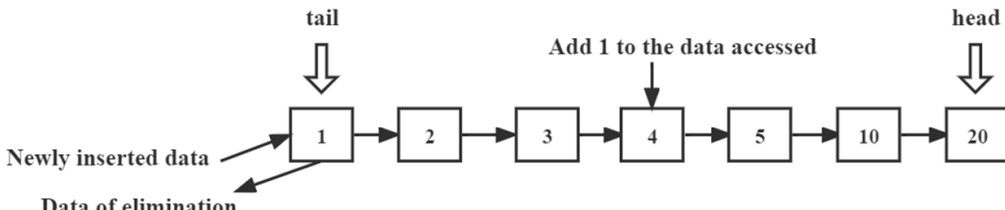

**Figure 3.** The idea of the LFU.

### 2.2.3. Size

The Size algorithm [28] is a representative algorithm based on the size of the data. When replacement occurs, larger data are deleted preferentially. This algorithm preferentially deletes large data and has high storage space utilization. However, it may cause hot data to move frequently, which reduces the hit rate and increases the access delay.

### 2.2.4. TCR

The TCR (Temperature Calculation Replacement) algorithm [29] takes into account the time interval and access frequency of data access. The calculation formula of the algorithm is given in Equation (1). $T_{t_n}$ indicates the temperature of the data at the time $t_n$. The cooling coefficient $\alpha$ is the change rate of the data temperature $T_{heat}$, denoting that the temperature has increased since the data were accessed.

$$T_{t_n} = T_{t_{n-1}} e^{-\alpha t_n - t_{n-1}} + T_{heat} * c, \begin{cases} \text{data are visited in } t_n, \ c = 1 \\ \text{data are not visited in } t_n, \ c = 0 \end{cases} \tag{1}$$

When replacement occurs, the algorithm preferentially deletes low-temperature data. It considers many factors, such as time interval and access frequency, and performs well. However, it needs to consider the size of the data, resulting in a waste of storage space. In addition, it wastes storage space by not taking into account the size of the data, and a lot of time is spent sorting the data.

### 2.3. Secondary Index

Index stores values of specific columns in the table and pointers to the addresses i the row [30–33]. Other fields corresponding to a record are not usually stored in the index, so a pointer is needed to find the history. Data structures commonly used to store indexes include B-Tree [34], hash index [35], R-Tree [36], bitmap index [37], etc. The time complexity of the B-tree is low. Its addition, deletion and changes in logarithmic time and the stored data are ordered. The hash index is based on a hash table. It only stores the corresponding hash value, and its structure is very compact. Its search speed is very fast. The R-Tree index is the extension of the B-Tree in the multidimensional index space and has high storage efficiency but low retrieval efficiency.

HBase uses Log Structure Merge Tree (LSM-Tree) to improve writing speed as an index structure [38–40]. LSM-Tree is a disk-based data structure that can significantly reduce the cost of disk traversal [41]. It stores recently used or frequently used data in memory and infrequently used data in hard disks, significantly reducing storage costs. As shown in Figure 4, it uses multiple small trees to store data. Its retrieval process is to build an ordered small tree in memory. As the amount of data grows, the data in memory is flushed to disk. However, it does not achieve a fast response because its retrieval result is obtained by traversing all the small trees [42,43].

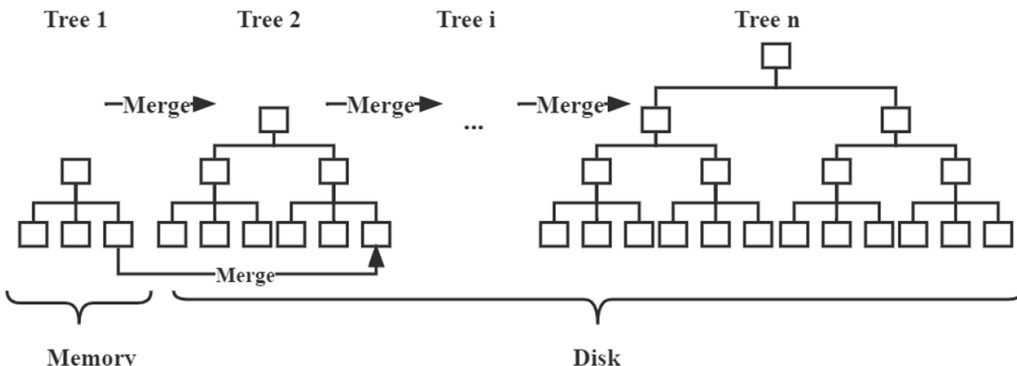

**Figure 4.** The structure of LSM-Tree.

Using secondary indexes is an effective method to support diversified queries in HBase. To date, many researchers have proposed a variety of HBase-based secondary index design methods [44–46], including a linear index, a double-layer index, and an inverted index.

Linear indexing achieves multidimensional indexing by mapping K-dimensional data to a one-dimensional space. This method is effective for processing high-dimensional spatial data but is not applicable to indexing other modal data [44]. The double-layer index matches the global index with the local index, thus reducing the number of query nodes and narrowing the query range from the high-level index to the low-level index. However, it requires maintaining two indexes and has a high write overhead. In addition, the two indexes require using different data structures and are very complex to implement [45]. The inverted index is the simplest multilevel index solution. Figure 5 shows the secondary index created based on the inverted index idea in HBase. The core idea is to use the index column in the main table as the key of the index table and the key of the main table as the value of the index table [46]. However, it cannot avoid querying the index table and the main table, resulting in high I/O overhead.

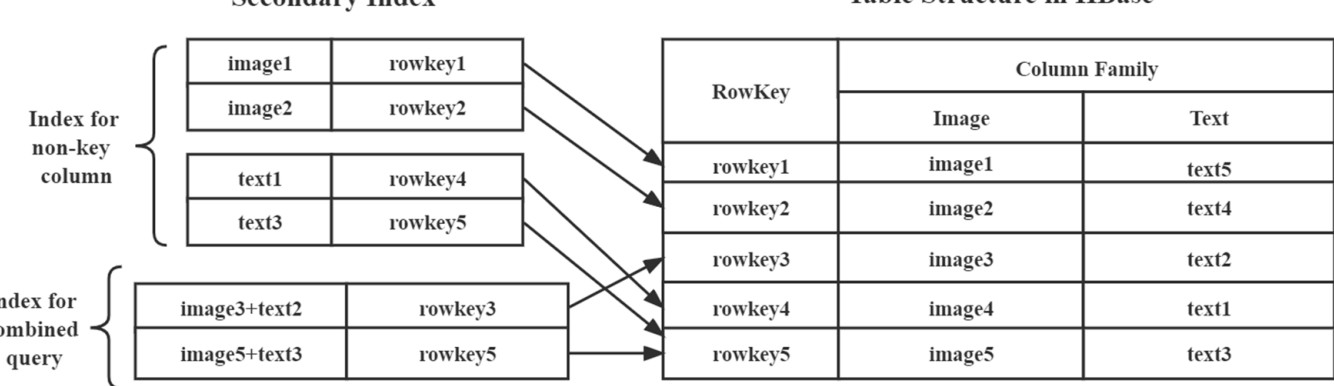

**Figure 5.** The secondary index in HBase.

*2.4. Summary*

As discussed in this section, the storage architecture of the HBase database was analyzed. It was found that HBase did not dynamically manage hot data, and the retrieval performance of non-primary keys was poor. Inspired by hot and cold management algorithms and secondary indexes, we proposed an HBase-based distributed storage and retrieval optimization model to optimize the retrieval scenario of data in the medical field.

## 3. Materials and Methods

### 3.1. Overview

HBase was employed to store medical data, where a column cluster stores one form of medical data. In order to improve the retrieval performance of the medical storage system, an HBase-based retrieval optimization model was proposed, and its design framework is depicted in Figure 6. The optimization module consists of four parts: the Access Request Management Module, the Temperature Marking Module, the Data Dynamically Management Module, and the Index Management Module. The Access Request Management Module manages access requests by analyzing the type of data to be retrieved. The Temperature Marking Module identifies the temperature of the data by analyzing data access records. The Data Dynamic Management Module dynamically manages the optimal storage medium for data by designing the algorithm SL-TCR. The Index Management Module, based on the improved secondary index strategy, can realize diversified retrieval of medical data.

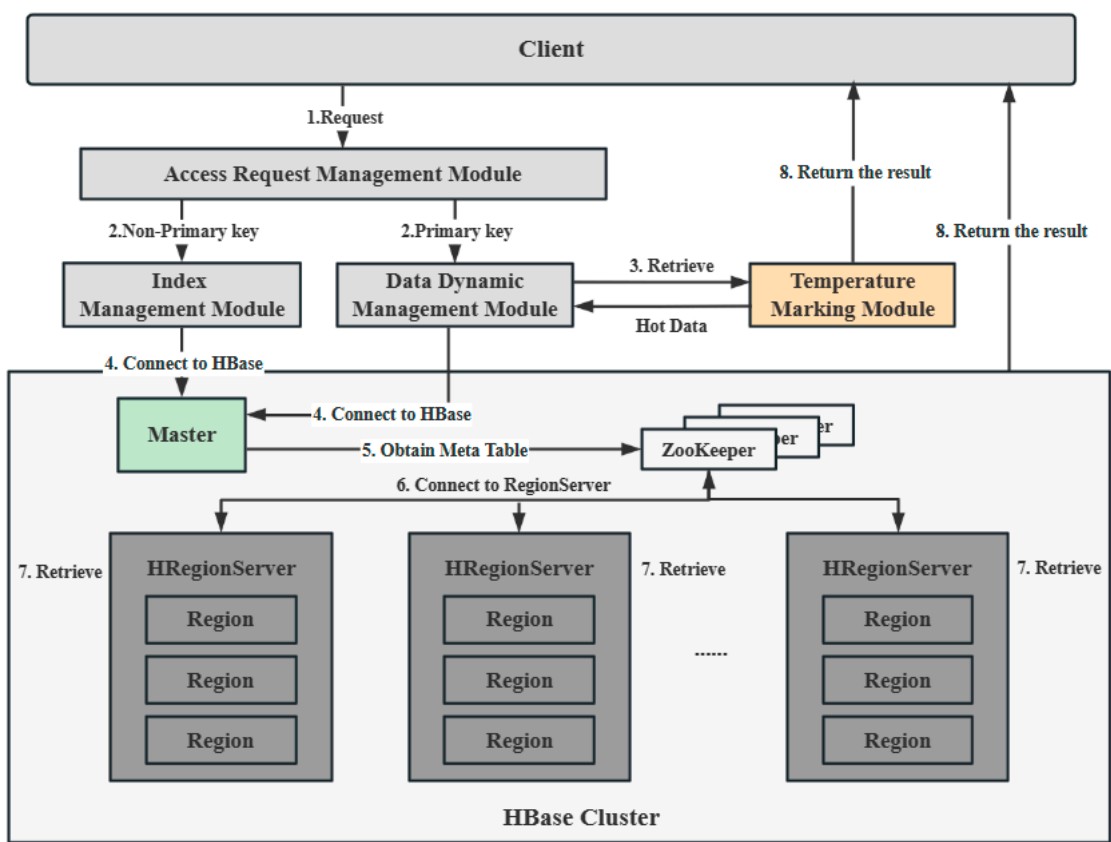

**Figure 6.** Overview of the model designed.

After introducing dynamic data management and index optimization strategies, the steps to retrieve medical data are as follows.

Step 1. The client sends a medical data access request to the Access Request Management Module.

Step 2. According to the retrieval keywords, the Access Request Management Module estimates whether it is a primary key. The request is passed to the Data Dynamic Management Module if it is. Otherwise, it is given to the Index Management Module (the retrieval process for non-primary keys is described in Section 3.4).

Step 3. The Data Dynamic Management Module determines whether the data to be retrieved is hot. If it is, the data in the Hot Area is retrieved, and go to Step 8. Otherwise, go to Step 4.

Step 4. Interact with the Meta Cache to read information about the RegionServer where the Meta table is located. If the Meta Cache does not match, connect to Zookeeper to obtain information about the RegionServer where the Meta table resides.

Step 5. Obtain the specific location of the Meta table, locate the RegionServer where the Meta table is situated, communicate with the node to obtain the Meta table, and write the Meta table metadata information to the Meta Cache.

Step 6. Interact with the Meta Table to read information about the RegionServer where the data to be retrieved is located, establish a connection with the node, and retrieve data in HBase. If no match is performed, go to Step 7. Otherwise, go to Step 8.

Step 7. An initial temperature is assigned to the retrieved data, and the algorithm SL-TCR is invoked to insert it into the Warm Area.

Step 8. Return the results to the Client.

*3.2. Temperature Marking Module*

By analyzing data access records in various specific medical scenarios, this paper finds that medical data has a relatively fixed access mode in different business scenarios, especially when specific data are accessed more frequently [47], called hot data. On the contrary, cold data are data that is accessed occasionally or will not be accessed in the future.

Different "measurement criteria" under the cold and hot degree of data will appear in different results. [48–50] The existing scheme usually uses the following three methods to identify the cold and hot degree of data: one is based on the sequence of data generation identification method, i.e., the earlier the data generated, the colder, the later the data generated, the hotter, usually using FIFO maintenance data insertion sequence; The second is the identification method based on data access frequency, i.e., the data with higher historical access frequency is hotter, and the data with lower access frequency is colder. Usually, the LFU algorithm can be used to maintain the sequence of data according to the historical access frequency. The third is the recognition method based on the data access sequence. That is, the more recently accessed data are hotter, and the earlier accessed data are colder. LRU algorithm is used to maintain data access to identify the degree of cold or hot data. However, these identification methods consider a single factor, and the identification effect of identifying the cold and hot degree of data simply according to the access time or frequency of data are relatively poor, which cannot truly represent the real cold and hot situation of data.

On this basis, a method of size–temperature computational recognition (S-TCR) is proposed in order to better identify the computational recognition of hot and cold data and take various factors into consideration. The method of S-TCR data cooling and heat labeling is to learn from Newton's cooling law and simulate the process of temperature change through exponential attenuation. As shown in Formula (2), Newton's cooling law proposes that an object with a high temperature in the physical environment will gradually cool down, and the temperature of the object will tend to the ambient temperature with the passage of time. Similarly, the temperature of the stored data decreases over time; when accessing data, it is similar to "warming" the data. The temperature of the data increases. In this way, we can acquire the temperature value of the data in the Hot Area at any time, and then sort the data according to the temperature value and define the K data with the lowest temperature as the cold data so as to realize the identification of hot and cold data.

$$T_t = (T_0 - H)e^{-kt} + H \tag{2}$$

where $T_t$ represents the current temperature of the object, H is the ambient temperature, and k is the proportional coefficient of the difference between the speed of temperature change in an object and the temperature of the surrounding environment.

The change law of objects in the physical environment affected by ambient temperature is slightly different from the change law of cold and hot degrees of data in data storage. In data storage, each datum is independent; the temperature of the data is not affected by other data or the storage media, but by the number of and access time of the data

itself. Therefore, if data are not accessed for a long time, its temperature will eventually be infinitely close to 0. That is to say, for data, its ambient temperature has no effect on its own temperature, thus the ambient temperature can be ignored when calculating the change in data temperature over time. Therefore, for the application scenario of measuring the cold and hot degree of data, Formula (2) is deformed, ignoring the influence of ambient temperature H, and variable $T_{heat}$ is added, namely, the "warming" amplitude of data after each visit. Formula (3) can be obtained:

$$T_{t_n} = T_{t_{n-1}} e^{-\alpha t_n - t_{n-1}} + T_{heat} * c \tag{3}$$

where $T_{t_n}$ indicates the temperature of the data at the time $t_n$, the cooling coefficient $\alpha$ is the change rate of the data temperature, $T_{heat}$ denotes the temperature increase since the data were accessed, and c represents whether the data are accessed at $t_n$. If so, it is 1. Otherwise, it is 0.

Medical data includes KB of text data and MB of image data, meaning that one also needs to consider the size of the data. At the same time, the log value of the data size is used to reduce the weight of the data block size, as to avoid large data blocks from being mislabeled and residing in high-cost media for a long time [51]. To sum up, the calculation formula of the S-TCR identification method is shown in Formula (4). Where Size denotes the size of the data.

$$T_{t_n} = T_{t_{n-1}} e^{-\alpha \lg(Size) t_n - t_{n-1}} + T_{heat} * c \tag{4}$$

The S-TCR method measures the degree of cooling and heating of data in data storage, specifically for the following three applications. (1) Data insertion: When the data are newly inserted, the ambient temperature of the data storage is taken as the initial temperature T0 of the data and assigned to the data; (2) Data access: When the data are accessed (Select, Update), the heat of the data increases. It is assumed that different access operations increase the temperature of the data equally, which is $T_{heat}$. Therefore, the temperature at the time when the data are accessed is the temperature obtained with time cooling, and then $T_{heat}$ is added; (3) Cold and the hot degrees of data: This method can calculate the real-time temperature of any data at any time and mark the cold and hot degrees of data. If you want to compare the cold and hot degrees of different data, you can directly compare the temperature values of the data. The data with a high temperature are relatively hot, while the data with a low temperature is rather cold.

The temperature model plays an important role in identifying hot and cold data. Through the exact temperature value, the temperature model realizes the quantification and identification of the cold and hot degrees of the data. Because the temperature model not only considers the influence of access frequency, time factor, and Size on the cold and hot degree of the data, it also uses the exponential calculation, thus, in the actual workload at any point in time, the temperature of any two data is different. It is more conducive to identifying the cold and hot degrees of data. In order to analyze the performance of the S-TCR method, the general properties of the S-TCR in quantifying the degree of cooling and heating of data are discussed.

In the following example, a variety of typical examples are selected. Formula 4 is used to calculate the real-time temperature of the data, assuming that the initial temperature of the data is the same, $T_0 = 30$, $T_{heat} = 2$, and the cooling rate of the data temperature is $\alpha = -0.05$.

(1) The temperature change in S-TCR with time was simulated only considering the access time. Data-1 was the data that had never been accessed. Data-2 refers to the data that wre frequently accessed in the first 100 s and never accessed in the last 200 s. Data-3 refers to the data that were never accessed in the first 200 s and frequently accessed in the following 100 s. Data-4 is the data that were never accessed in the first 280 s and frequently accessed in the second 20 s. The temperature changes of four kinds of data over time are shown in Figure 7.

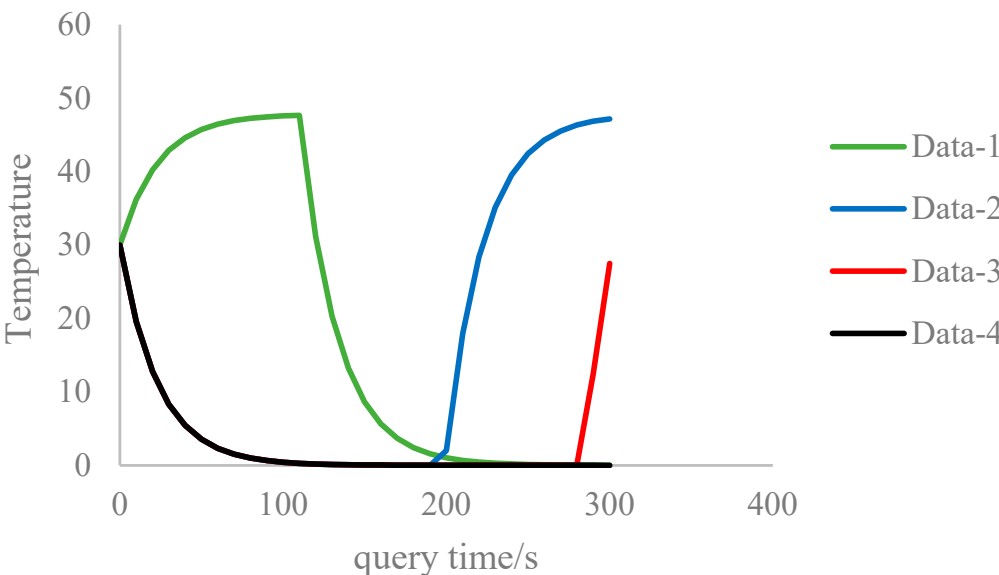

**Figure 7.** Influence of access time on temperature in S-TCR method.

As shown in Figure 7, when Size, access frequency, and other conditions are consistent, the hot data frequently accessed in the early stage will gradually cool down to cold data due to the cooling mechanism. When other conditions, such as Size and access frequency, are consistent, the frequently accessed data in the later period will gradually heat up to become hot data. It shows that this method pays more attention to the recent thermal data and avoids the pollution of the Hot Area.

(2) The temperature change in S-TCR with time was simulated only considering the access frequency. Data-1, Data-2, and Data-3 are the data that were never accessed in the first 280 s but were frequently accessed in the later 20 s. In addition, Data-1 was the control group; Data-2 was accessed twice as often as Data-1; Data-3 was accessed three times as often as Data-1. The temperature changes of three kinds of data over time are shown in Figure 8.

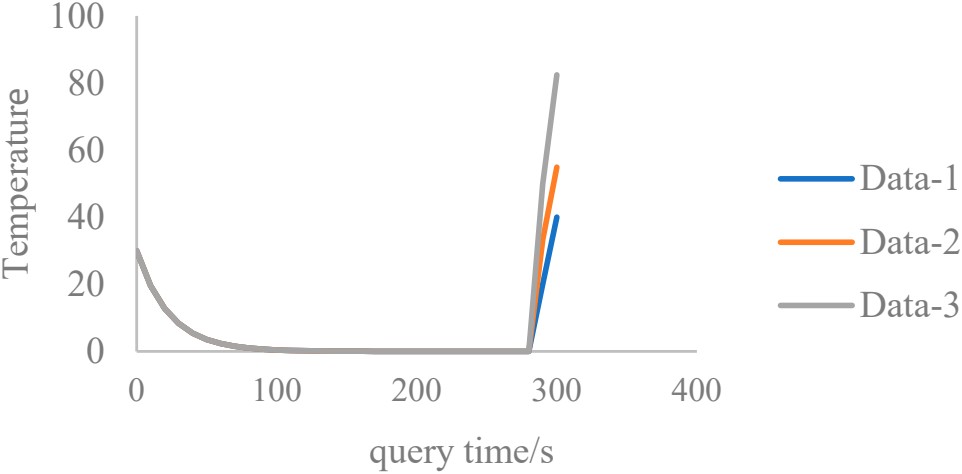

**Figure 8.** Influence of access frequency on temperature in S-TCR method.

As shown in Figure 8, when the number of accesses is different, the temperature of the later accessed data may not be high. When conditions such as Size and access time are consistent, the higher the access frequency in the S-TCR method, the higher the temperature.

(3)　Temperature changes of S-TCR over time were simulated only considering data Size. Data-1, Data-2, and Data-3 are the data that were never accessed in the first 280 s but were frequently accessed in the later 20 s. In addition, Data-1 was the control group; The Size of data 2 was 100 times that of Data-1. Data-2 was 1000 times larger than Data-1. The temperature changes of three kinds of data over time are shown in Figure 9.

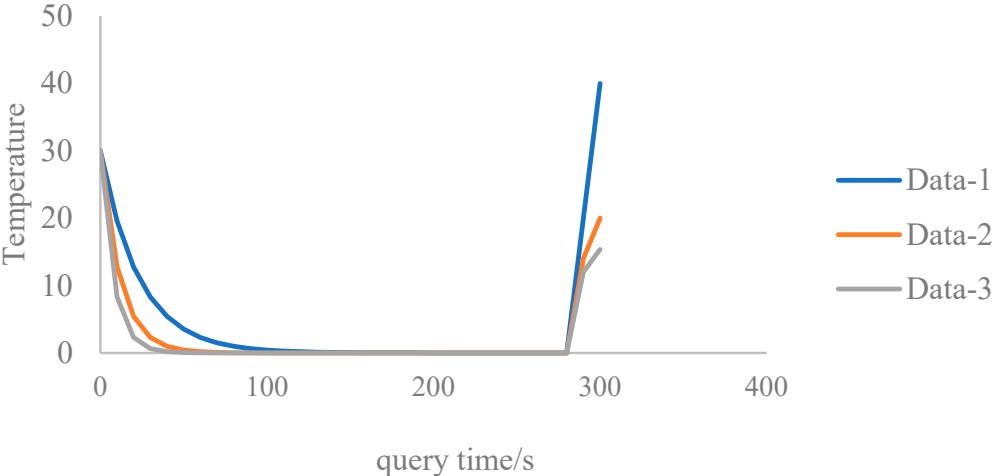

**Figure 9.** Influence of Size on temperature in S-TCR method.

As shown in Figure 9, we found that data with larger sizes in the S-TCR method cooled down faster and warmed up faster. In addition, we found that the cooling and warming amplitude of the data caused by Size was reasonable despite the large difference in the size of the data, which also showed that formula 4 was reasonable for the treatment of Size.

In summary, the S-TCR identification method comprehensively considers the access frequency, access time, and size of multi-modal medical data. It realizes the temperature identification of any data at any time, avoiding the performance limitation caused by a single factor of LRU, LFU, Size, and other algorithms. Similarly, it is more suitable for multi-mode medical data than the TCR algorithm. Therefore, the S-TCR identification method is in line with our design expectations and can effectively identify hot and cold medical data.

*3.3. Dynamic Management Module of Data*

Access to data is dynamic, and the storage capacity of high-cost storage media is limited. Therefore, it is necessary to design cold and hot data management modules and realize a dynamic data management model based on data temperature to improve access performance. The module uses the HBase database as the Cold Area and memory as the Hot Area. It mainly implements the following three functions: (1) Data insertion: when newly accessed data are not in the Hot Area, an initial temperature $T_0$ is assigned to the number and inserted into the Hot Area; (2) Data query: when an access request arrives, the data will be retrieved in the Hot Area, and the result will be returned. If the search keyword does not exist, the request will be returned; (3) Data replacement: when the Hot Area reaches the threshold, an appropriate replacement algorithm is selected to delete the Cold Data that have been cooled in the Hot Area in bulk.

The performance of the data replacement algorithm is analyzed below. Since we have proposed the temperature identification method S-TCR in Section 3.2, we naturally thought of using it to replace the cooled data in the Hot Area. By calculating the temperature of all the data in the Hot Area, we used the sorting algorithm to select the K coldest data. Although this method is simple and easy to implement, the cost is very high, and the cost of data replacement is very high. However, the LRU algorithm can be completed only in O (1) time complexity. According to statistics, under 100 K visits, the time spent by the S-TCR algorithm is about three times that of the LRU algorithm [29]. Compared with LRU,

the hit rate of the S-TCR model can be increased by about 1.5 times, which indicates that the S-TCR method is more accurate in identifying the degree of cold and hot data. While the LRU algorithm has a lower time cost, the two can be combined to take advantage of each other. The SL-TCR (Size & LRU–temperature Calculation Recognition) algorithm is proposed to replace hot and cold data.

The Hot Area is divided into Hot Area and a Warm Area by the SL-TCR algorithm. warm data refers to cooler hot data, and the rate of Hot Area and Warm Area is 3:1. The SL-TCR algorithm uses the LRU algorithm to manage data in Hot Area dynamically, and the S-TCI temperature recognition method uses a sorting algorithm to dynamically manage data in Warm Area, which avoids traversing all the cached data, reduces the time overhead of the algorithm, and improves the performance of the algorithm. Figure 10 describes the replacement idea of the SL-TCR algorithm. In the beginning, both the Hot Area and the Warm Area are empty. As time goes by, data are accessed continuously. The recently accessed data are recorded as warm data, and the warm data are moved into the Warm Area. If the Warm Area is full, the heated data in the Warm Area, namely, hot data, are transferred into the Hot Area, and the warm data are moved into the Warm Area. If the Hot Area and Warm Area are both full, the cooled Data in the Hot Area and Warm Area are deleted, and the warm data are moved into the Warm Area.

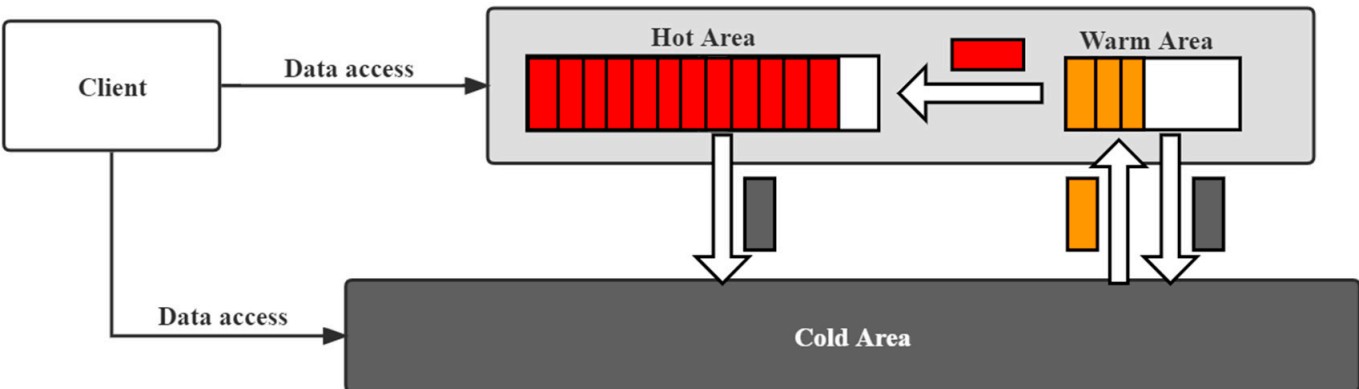

**Figure 10.** Data dynamic management model data replacement idea.

The algorithm SL-TCR (Size & LRU-temperature Calculation Recognition) is used to manage hot and cold data. This algorithm uses the LRU algorithm to dynamically manage data in Hot Area and the S-TCR temperature recognition method and sorting algorithm to manage data in Warm Area dynamically. The data block, designed by SL-TCR for storing data, is composed of five parts: key, value, T, t, and size. They represent, respectively, the key of the data, the value of that key, the temperature of the data block, the timestamp when the temperature was last calculated, and the size of the data block.

The process of the SL-TCR algorithm proposed in this paper is shown in Algorithm 1. Lines 2 through 4 indicate that if the Warm Area is not full, new data are inserted directly into the Warm Area. Lines 5 through 11 state that if the Warm Area is full and the Hot Area is not, then the heated warm data from the Warm Area will be moved to the Hot Area. Lines 12 through 17 suggest deleting the warm, cooled data in the Warm Area when both Warm and Hot Areas are full. The temperature of the KTH data is denoted as Cold. Data blocks with lower temperatures than Cold are deleted from the Hot Area. Line 18 indicates inserting data into the Warm Area.

**Algorithm 1. The process of algorithm SL-TCR**

**Input:** key, value, T, t, size, threshold1, threshold2
**Output**
1.  data = **new** Node(key, value, T, t, size)
2.  **if** WarmArea.size() < threshold1 **then**
3.      WarmArea.put(data);
4.  **end**
5.  **else if** WarmArea.size() $\geq$ threshold1 **then**
6.      update_Temperature();
7.      HotCold $\leftarrow$ sort(stcr);
8.      **if** ltcr.size < threshold2 **then**
9.          WarmArea.remove(Hot);
10.         HotArea.put(Hot);
11.     **end**
12.     **else**
13.         **for** i $\leftarrow$ 0 **to** k **do**
14.             Cold $\leftarrow$ WarmArea.remove();
15.         **end**
16.         HotArea.remove(node.T < Cold.T);
17.     **end**
18.     WarmArea.put(data);
19. **end**

### 3.4. Index Management Module

When searching in HBase for data by non-primary key, the result can be obtained only by scanning the entire table and filtering the data that does not meet the search criteria. However, scanning tables with hundreds of millions of records will take up a lot of resources. Therefore, we need to design a secondary index for HBase tables to avoid the time consumption in retrieving non-primary keys. The secondary index stores the mapping between index columns and keys and is a common and efficient solution for searching for non-primary keys. It searches the RowKey through the secondary index, and the corresponding complete data can then be searched via the RowKey.

Further, two optimization strategies for the secondary index were proposed in the present study to reduce the retrieval time and improve the retrieval efficiency of the index. Firstly, Bloom Filter was used to optimize the performance of the secondary index. BF is an effective method to judge whether an element w exists in set A, especially when the number of elements in A is very large and the amount of data far exceeds the memory space of the machine [52–55]. Hence, BF is used to discover non-existent search keywords to avoid unnecessary time overhead generated by extreme I/O. The BF mapping index keyword is shown in Figure 11, and its idea is described as follows:

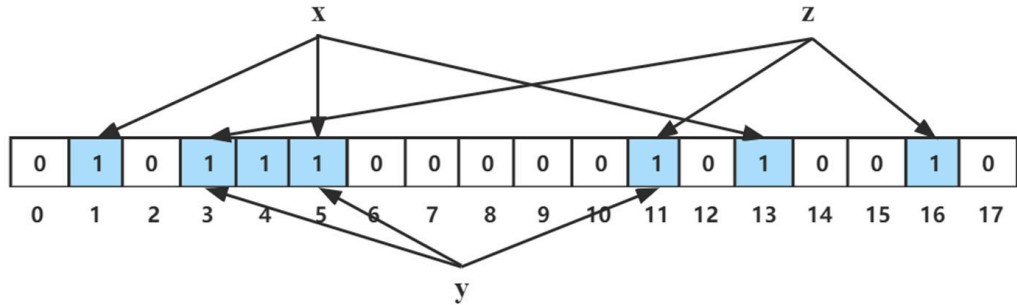

**Figure 11.** The process of hash mapping of BF.

Step 1. Create an array A of length n with elements of 0 or 1.

Step 2. Each element w of A is initially set to 0.

Step 3. For each keyword w of the index, conduct k hashes, the i'th hashes modulo N, generate the mappings, and set them to 1 (for example, the x, y, and z are mapped to 1, 5, 13, 4, 11, 16 and 3, 5, 11 by hash function).

Step 4. When a non-primary key retrieval occurs, the mapping values of the search keywords are obtained through k hashes, and the values of their corresponding bits are obtained in the A.

If any are not 1, the keyword will fail to be matched. This situation means that the query result does not exist, and the query will be filtered out (for example, the w is, respectively, mapped to bit 0, bit 3, and bit 15 because there are non-1 bits in the resulting mapping value, thus it is determined that w must not be in the A).

Step 5. If the keyword is successfully matched, continue to retrieve.

It is worth noting that there is a case where the match is successful, but the element does not exist in set A. Such a miscalculation is called a False Positive. For example, the mapping values of t is 5, 11, and 13, respectively, and these positions are all 1 in A. In fact, t is not in A. As indicated in that study, let $F_b$ be the rate of the False Positive. $F_b$ can be expressed as Equation 5, where m represents the bit number of BF, k represents the number of hash functions, and n represents the number of elements in the set [52]. Due to the low $F_b$, it is assumed in this paper that there is no keyword misjudgment.

$$F_b = [1 - \left(1 - \frac{1}{m}\right)^{kn}]^k \approx \left(1 - e^{\frac{-kn}{m}}\right)^k \tag{5}$$

Secondly, the storage location of the secondary index is designed to reduce the time cost of the index based on an inverted index. Generally, if an HBase table is not large, a Region is used to store the table and a RegionServer [56] is used to monitor the table. As the data size increases, the Region may be split and monitored by multiple RegionServers [56]. Due to the large size of the medical data, secondary indexes and primary data are likely to be in different Regions and monitored by different RegionServers [57], resulting in non-primary key retrieval requests requiring four I/O operations to acquire results. Specific operations are as follows: (1) the client queries the index table based on the retrieval keyword; (2) acquire the RowKey of the main table and return it to the client; (3) the client queries the main table according to the RowKey; (4) the retrieval result is returned to the client. Obviously, multiple communications with the RegionServer increases the time overhead and leads to low retrieval efficiency. Therefore, it is very necessary to optimize the storage location of secondary indexes and reduce the number of I/O communication to improve the retrieval efficiency of secondary indexes.

The retrieval performance can be improved by reducing the number of I/O operations. Suppose the main table and index reside on the same RegionServer and are run on it using the coprocessor provided by HBase. In that case, the query requires only 2 I/O operations: (1) query the index table and obtain the RowKey according to the data to be queried, then query the main table in accordance with the RowKey; (2) return the result to the client. Therefore, for optimization, we should consider how to host the main and index tables on the same RegionServer. For the purpose of achieving the goal, the field of RowKey is designed as demonstrated in Figure 12. The RowKey of the secondary index consists of 3 fields: (1) 0 to 8 bits indicate the start key of the Region where the data in the main table resides. The search of RowKey follows the rule of the leftmost prefix. Therefore, the index table and the main table are in the same RegionServer; (2) 9 to 16 is the index name that uniquely identifies the index; (3) 17 to 17 + m bits to ensure the uniqueness of RowKey. m is the minimum number of bytes required to provide the uniqueness of RowKey.

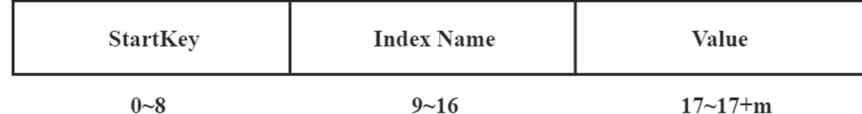

| StartKey | Index Name | Value |
|----------|------------|-------|
| 0~8 | 9~16 | 17~17+m |

**Figure 12.** The RowKey of designed secondary index.

After the introduction of the secondary index optimization strategy, the retrieval steps are shown in Figure 13.

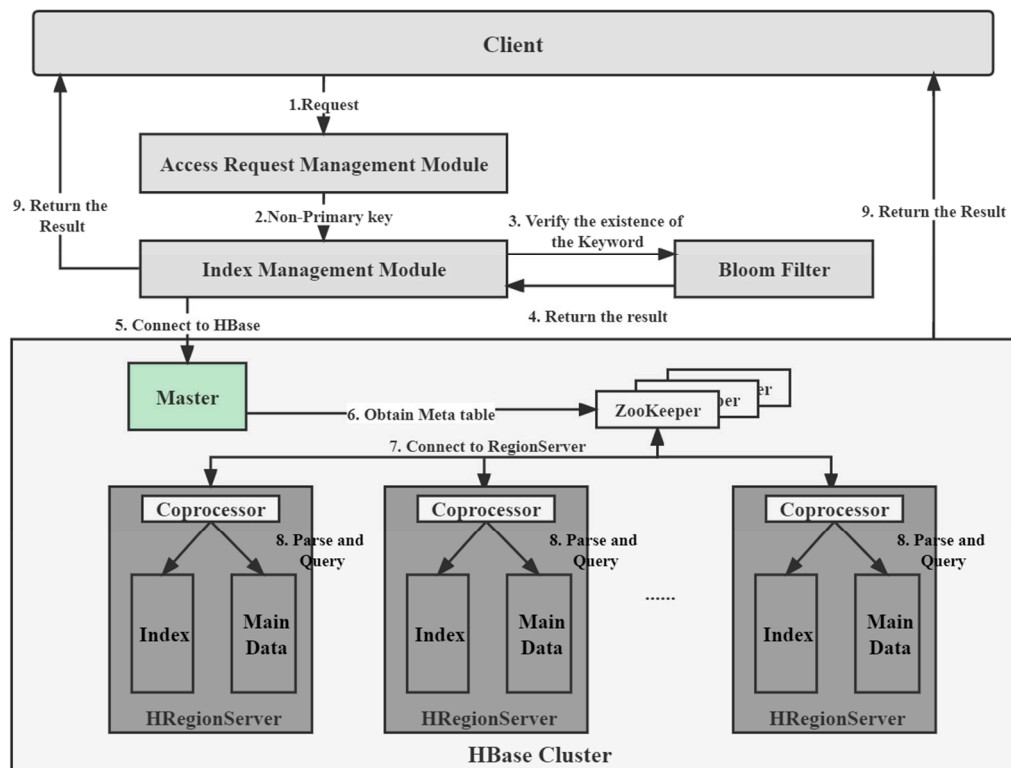

**Figure 13.** The retrieval steps of the secondary index.

Step 1. The client sends a medical retrieval request to the Access Request Management Module.

Step 2. The Access Request Management Module determines whether the keyword is a primary key. If it is a non-primary key query, the request will be passed to the Index Management Module.

Step 3. The BF determines whether the keyword matches successfully.

Step 4. If the match is successful, go to Step 5. Otherwise, go to Step 9.

Step 5. Connect to HBase.

Step 6. Interact with the Meta Cache to read information from the RegionServer about where the Meta table is. Communicate with the RegionServer where the Meta table is located to obtain the Meta table.

Step 7. Communicate with the RegionServer about where the data are located.

Step 8. The coprocessor receives the request and then parses and queries it. Afterwards, query the index table to receive RowKey. Then, the main table is queried by RowKey.

Step 9. Return the results to the client.

## 4. Experiments and Results

In the experiments, 15 servers were employed to build a Hadoop-distributed cluster. Hadoop 3.1.1, Ubuntu 16.04, a Core i7-10700 CPU, 32 GB RAM, and the HBase software version 1.4.13 were used in the cluster. According to the model designed in this paper, the MIMIC-IV dataset [58] was stored in HBase. MIMIV-IV is one of the commonly

used international public healthcare data sets, which contains data in three modes: two-dimensional tabular data, text-based diagnostic reports, and image data. The performance of the dynamic management model was verified by comparing the hit ratio and access latency. Moreover, the performance of the index optimization strategy was verified by comparing the access delay of non-primary key retrieval under different models.

### 4.1. Performance of Dynamic Management Model

This experiment used memory and the HBase database as the Hot Area and the Cold Area, respectively. The hit rate of SL-TCR, LRU [26], and S-TCR algorithms in Hot Areas were compared to verify the performance of the proposed SL-TCR algorithm through the medical data query experiment. The specific settings of the experiment were as follows: the initial query times were 2000, and the query times were increased by 2000. The number of batch deletions K was 5%, and the temperature attenuation coefficient $\alpha$ was 0.01. The comparison results of the three algorithms are shown in Figure 14. In addition, the performance of the model in this paper was confirmed by comparing the access latency of the original HBase, S-TCR, and SL-TCR models. The access delay experimental results of the three models are shown in Figure 15.

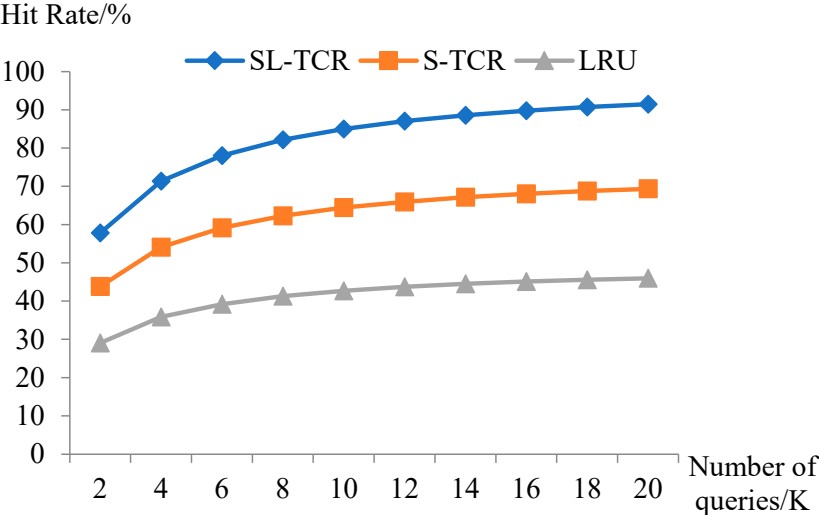

**Figure 14.** Hit rate comparison of three algorithms.

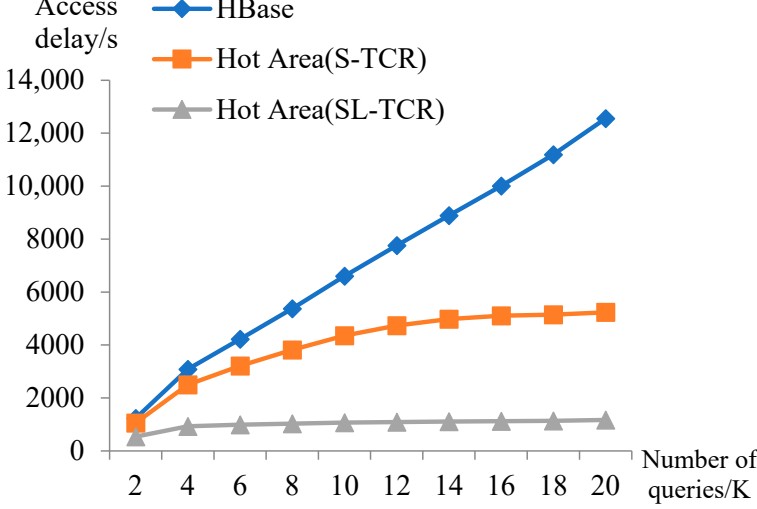

**Figure 15.** Access latency comparison of three models.

According to the experimental results in Figure 14, the following conclusions can be drawn: First, the hit rate of the three algorithms increased with the increase in the number of queries. It showed that the algorithm had good adaptability and stability to historical access records. Second, the hit rate of SL-TCR algorithm was higher than that of the LRU and S-TCR algorithm. The hit rate of SL-TCR was 28.76–45.51% higher than that of LRU. This can be attributed to the fact that the SL-TCR algorithm took more factors into account than LRU, such as access frequency and size. As Warm Area was introduced into SL-TCR, the mining of medical hot spot data was more effective, and its hit rate was 13.98–22.11% higher than S-TCR. In addition, compared with S-TCR and SL-TCR, the higher the hit rate of the algorithm, the lower the time cost of the algorithm. If the number of access times was W, the average hit ratio was r, the number of batch deletes was K, and the number of data stored in the Hot Area was S, the number of data replacement times was $(W(1-r)-S)/K$. It can be seen that the higher the hit rate of the algorithm, the fewer the number of data replacement and the lower the time cost of the algorithm. For example, if the Hot Area could hold 400 pieces of data, the hit rate of access using S-TCR algorithm was 60%, K = 40, and about 1000 culling will occur. The hit rate of SL-TCR algorithm access was 80%, K = 40, and about 500 eliminations occur. In addition, because SL-TCR algorithm avoids traversing all cached data, the time cost of SL-TCR algorithm was much lower than that of S-TCR algorithm. In addition, the hit rate of SL-TCR model was about two times higher than that of LRU. In conclusion, the experimental results show that SL-TCR algorithm realized the complementary advantages of the two algorithms, which is more accurate in identifying the degree of cold and hot data, and the algorithm also had lower time cost.

According to the experimental results in Figure 15, queries on the Hot Area storage model have lower access latency than those on HBase. This is because the dynamic management model improves the utilization rate of the hotspot area and proves that the dynamic management model can optimize the access performance of the system. Second, the effect of SL-TCR algorithm is more significant than that of S-TCR algorithm. The access latency of S-TCR ranges from 49.08% to 77.61%, lower than that of HBase. The access latency of SL-TCR ranges from 56.21% to 90.66%, lower than that of HBase, because the Warm Area is introduced in SL-TCR to reduce the amount of data sorting and time overhead.

In summary, the dynamic data management model based on the SL-TCR algorithm greatly improves the retrieval efficiency of the original HBase model, improves the accuracy of hot data identification, optimizes the algorithm performance compared with LRU algorithm and S-TCR, and can greatly improve the retrieval performance of HBase-based medical storage systems.

### 4.2. Performance of Secondary Index

Medical data were queried to verify the performance of the proposed secondary index strategy. Access delay was the evaluation index for this experiment. Four groups of comparison experiments were set up: the original HBase system, the system using the adding Bloom Filter, the system using the index storage location optimization, and the system using both optimization strategies. Other settings were as follows: the initial query times were 200, and the query times were increased by 200. The experimental results are demonstrated in Table 1.

**Table 1.** Performance of secondary index optimization strategies.

| Number of Queries | HBase | Bloom Filter | Same RegionServer | Ours |
|---|---|---|---|---|
| 200 | 332.78 | 292.988 | 324.5 | 279.1359 |
| 400 | 685.27 | 573.3687 | 658.03 | 552.9814 |
| 600 | 1027.73 | 822.6875 | 965.95 | 774.67 |
| 800 | 1358.72 | 1050.404 | 1279.28 | 985.3437 |
| 1000 | 1704.06 | 1290.018 | 1583.52 | 1179.21 |

According to the experimental results of the secondary l index optimization strategies obtained in Table 1, the following conclusions can be acquired. First, the two optimization strategies can improve the speed of non-primary key retrieval. Adding Bloom Filter reduced access delay by 12.0–26.6% compared to the HBase. This can be attributed to avoiding the I/O overhead of non-existent keywords to be retrieved. Second, compared with the HBase, the system with the index and main data on the same RegionServer had reduced access latency by 3.4% to 7.1%. This can be explained by the fact that this strategy reduced the number of I/O operations in a single retrieval from 4 to 2. Third, using both optimization strategies simultaneously reduced the access delay by 16.1–30.8%. Since the two optimization strategies have different optimization directions and do not interfere with each other, the simultaneous use of both optimization strategies was better than using only one optimization strategy. Consequently, it was demonstrated that the optimized secondary index optimization strategies proposed in this study improve the performance of non-primary key retrieval and can meet the objectives of fast and diversified medical data retrieval.

## 5. Conclusions

An HBase-based distributed storage and retrieval optimization model for medical data was proposed, and a retrieval optimization model based on dynamic management of the temperature of data and the improved secondary index was implemented. The dynamic management of data was introduced to identify the temperature of data, and data with different temperatures were stored in the corresponding areas, which can make full use of high-cost media and speed up retrieval. The improved secondary index uses a Bloom Filter to filter non-existent keywords and the designed RowKey to optimize index storage location, reducing I/O overhead and improving the retrieval performance of non-primary keys. The comparison with the original HBase system proves that the proposed optimization strategy can give full play to the advantages of the storage model, greatly reduce access latency, and meet the access requirements of frequent interaction of hot data and diversified queries in the medical service.

With the increase in the amount of medical data and the development of storage technology, HBase-based retrieval optimization will continue to be a research hotspot in the future under the retrieval scenario of massive medical data. In order to solve this problem, we hope to conduct further in-depth research on the following aspects: (1) Integrate the S-TCR method and SL-TCR algorithm into an open-source system to verify the model effect at the system level. (2) How to apply the optimized two-level strategy proposed in this paper to the multi-field query and range query is also one of the key points to be studied in the future.

**Author Contributions:** Conceptualization, C.Z. and Z.L.; Formal analysis, H.W.; Funding acquisition, C.Z.; Investigation, Z.L.; Methodology, Z.L. and Z.F.; Project administration, C.Z. and B.Z.; Resources, Z.F.; Software, M.Z.; Validation, H.W.; Visualization, M.Z.; Writing—original draft, Y.X.; Writing—review and editing, Z.L. and Y.X. All authors have read and agreed to the published version of the manuscript.

**Funding:** This research was funded by the National Key R&D Program of China, grant number 2018AAA0102100. This research was financed in part by the International Science and Technology Innovation Joint Base of Machine Vision and Medical Image Processing in Hunan Province, grant number 2021CB1013. This research was financed in part by the National Natural Science Foundation of China, grant number 61902434. This research was financed in part by the Natural Science Foundation of Hunan Province of China, grant numbers 2019JJ50826, 2022JJ30762. This research was financed in part by the Key Research and Development Program of Hunan Province, grant numbers 2022SK2054.

**Institutional Review Board Statement:** Not applicable.

**Informed Consent Statement:** Not applicable.

**Data Availability Statement:** MIMIC-IV Database dataset: https://mimic.mit.edu/docs/gettingstarted, accessed on 1 December 2022.

**Acknowledgments:** This work was financed in part by the National Key R&D Program of China and funded by 2018AAA0102100. This work was financed in part by the International Science and Technology Innovation Joint Base of Machine Vision and Medical Image Processing in Hunan Province and funded by 2021CB1013. This work was financed in part by the National Natural Science Foundation of China and funded by 61902434. This work was financed in part by the Natural Science Foundation of Hunan Province of China and funded by 2019JJ50826 and 2022JJ30762. This work was financed in part by the Key Research and Development Program of Hunan Province and funded by 2022SK2054.

**Conflicts of Interest:** The authors declare no conflict of interest.

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
