# Peer review of "An HBase-Based Optimization Model for Distributed Medical Data Storage and Retrieval"

_electronics, doi:10.3390/electronics12040987_

Round 1

Reviewer 1 Report

The authors improve the efficiency of HBase in medical data management in two ways: 1) In order  to improve the retrieval speed , they propose new algorithms to identify the frequently accessed hot data and dynamically manage the data storage medium 2) In  order to improve the search performance, they propose a secondary index strategy to  reduce I/O overhead and optimize the search performance of non-primary key indexes. They also give interesting experimental results demonstrated the efficiency of the above mentioned new algorithms. Their references are sufficient. The English language and style are good enough.

I have few comments:

1. Since they referred to medical data, I feel that they have to specify exactly and in more details their characteristics (of medical data) and the new imposed requirements, concerning their processing, management and store.

2. Equation (2) that is one of the main contributions of the paper must be explained better.

3. Figure 6: Data Dynamically or Data Dynamic ?

4. Algorithm 1, line 5: It seems to me that you have to use <else if> command instead of <else>.

5. There are some typos, eg:

            Line 288: is mapped to … (what?).

            Line 289: delete dot (.)

            Line 295: “the of t is…” , incomplete sentence, what is t?

            Line 296:”Fb” or F(pointer)b?

            Line 302: “Secondly”, but I cannot find the … “firstly”

            Line 306: “different” RegionServer (?)

            Line 307: 4 I/O , better four I/O

6. Lines 304-306: I feel that have to be explained better compared to lines 307-311

7. Finally, I think the authors have to consider and refer some related future work.

Reviewer 2 Report

The manuscript introduces an improved model for medical data storage and retrieval based on a map-reduce approach. It distinguishes between hot and cold data to optimize data access. 

The draft is well-structured and readable.

Comments:

- how exactly is the dynamic modelled in the data base accesses? How does the results change dependent on it? A more detailed study is required here in my opinion.

- more details about what is hot and cold data and the distribution within the whole data set (also over time) would be important to understand the influence on the performance. 

- results for a higher number of queries should be added and the impact on performance should be stated

- table 1: please mention in the tabel or caption again what performance means here extactly, this makes it easier to read

- an integration in the database system is missing as mentioned in the conclusion, are the results valid then for the real system?

In summary, I suggest a major revision of the manuscript and some more results and explanations have to be added as mentioned in the remarks to publish it.
